# Impact of COVID-19 Stay-at-Home Restrictions on Employment Status, Physical Activity, and Sedentary Behavior

**DOI:** 10.3390/ijerph182211935

**Published:** 2021-11-13

**Authors:** Cheryl A. Howe, Riley J. Corrigan, Fernanda Rocha de Faria, Zoe Johanni, Paul Chase, Angela R. Hillman

**Affiliations:** 1Applied Health Sciences and Wellness, Ohio University, Athens, OH 45701, USA; zj760116@ohio.edu (Z.J.); chasep@ohio.edu (P.C.); hillman@ohio.edu (A.R.H.); 2Honors Tutorial College, Ohio University, Athens, OH 45701, USA; rc401218@ohio.edu; 3Federal Institute of Education, Science and Technology of Triângulo Mineiro, Ituiutaba 38305-200, Brazil; frfaria.ef@gmail.com

**Keywords:** physical activity, sedentary behavior, exercise barriers, coronavirus

## Abstract

Background. North Americans report insufficient moderate-to-vigorous physical activity (MVPA) and ample sedentary behaviors (SBs), suggesting possible barriers to an active lifestyle. This study compared self-reported MVPA and SB before and during COVID-19 “Stay-at-Home” restrictions as a potential barrier across North America. Methods: Questionnaires were distributed from 21 April to 9 May 2020. ANOVAs compared data overall and by group (age, sex, race, income, education, employment status). Results: During restrictions, 51.4% (*n* = 687) of the 1336 responses (991 female, 1187 Caucasian, 634 18–29 years) shifted to work from home and 12.1% (*n* = 162) lost their job. Overall, during restrictions, 8.3% (*n* = 110) fewer reported work-related MVPA (−178.6 ± 20.9 min/week). Similarly, 28.0% (*n* = 374) fewer reported travel-related MVPA, especially females and younger age groups. While the 7.3% (*n* = 98) fewer reporting recreational MVPA was not statistically significant (−30.4 ± 11.5 min/week), there was an increase in SB (+94.9 ± 4.1 min/week) in all groups, except the oldest age group (70+ years). Locomotive activities and fitness class remained the predominant MVPA mode. Of those reportedly using facilities (68%; *n* = 709) before COVID, 31.3% (*n* = 418) would not return due to it “being unsafe”. Conclusion: While barriers related to pandemic restrictions had a negative short-term impact on MVPA and SB in North America, the long-term impact is unknown.

## 1. Introduction

Recent analysis of the 2003–2006 National Health and Nutrition Examination Survey (NHANES) data by Saint-Maurice (2020) found that only 50.7% of adults over 40 years reached a minimum of 8000 steps/day, which declines with age [1]. More recent data from the same national survey (2006–2017) found that physical activity (PA) levels have not improved and that disparities exist across age, sex, racial/ethnic minorities, and income levels [2]. It is well known that PA is negatively associated with chronic disease and all-cause mortality, but this association is not encouraging the population to become more physically active to promote health [3]. During times of global health crises, such as the Coronavirus Disease 2019 (COVID-19) pandemic, it is unclear if the worldwide emphasis on encouraging the betterment of health and wellness will awaken people to the need to adhere to an active lifestyle. Additionally, it is unclear if the restrictions imposed on society associated with reducing COVID-19’s spread will hinder this need. 

An integration of the social-cognitive approach, where perceived barriers to PA stem from a combination of internal or external sources [4], and the theory of planned behavior, which focusses on how perceived risks and benefits can affect PA participation [5], was used to assess changes in PA behavior. External barriers come in the form of time constraints and lack of facility access, while internal barriers include lack of motivation or energy, lack of social support, and perceived incompetence [6]. The environment also plays an important role in promoting or hindering PA participation. For example, a busy parent may have high demands from their family and job and might have little time or be too exhausted to carve out “me time” for being physically active, especially between their own work hours and shuttling children between various extracurricular commitments. These scenarios are all too common in developed countries where financial security and child experiences are valued, creating a lack of time and motivation to be physically active [6]. In times of pandemic restrictions, these and other unforeseen factors may strengthen the barriers to PA. 

With the “Stay-at-Home” restrictions mandating closure of fitness facilities, thus reducing PA opportunities, the theory of planned behavior framework was used to assess psychological factors that affect PA behavior [5,7]. Specifically, participating in PA within an enclosed fitness facility or in large groups amidst this pandemic may elevate the perceived risk due to the threat of COVID-19 exposure, which may overshadow the perceived benefits of PA participation. Therefore, this could force either a reduction in PA or a change in PA mode, even as these facilities begin to reopen.

During a global pandemic, environments for home and work can change drastically. “Stay-at-Home” restrictions designed to reduce the pandemic’s spread have changed the demands on the pre-pandemic lifestyle. Travel to and from work, school, and other daily activities used to contribute significantly to daily PA participation [8,9]. For those who typically drive to such activities, the switch to staying at home may create more time during the day to be used for recreational PA. However, overall, PA may decrease if travel-related PA is not replaced with home-based activity. In addition, pandemic restrictions often result in cancellations of youth sports and other family demands. This, in turn, could promote more family-centered PA. However, it has also been shown that increased family stressors can negatively influence family activity, especially in times of pandemic. Prime et al. (2020) found that times of quarantine added stressors, such as financial insecurity, increased childcare needs, and loss of routine, to explain why PA and self-care priorities may be put on the backburner [10]. As PA has been shown to help reduce stress [11], an increase in PA could be used as a coping mechanism to ease these added stressors. As indicated in a recent article by Napoli et al. (2021), lockdown or “Stay-at-Home” orders significantly alter the environment, personal behaviors (i.e., screen use and dietary habit), and psychological states because of the closure of schools, offices, and industries and bans on public gatherings, thus adding additional stress and increased perceived barriers to PA [12]. Additionally, Mertens et al. (2020) reported that the COVID-19 virus pandemic triggered significant fear related to worry, uncertainty, health anxiety, and concern for loved ones in more than 46% of adults questioned in an online survey (N = 439; 16–80 years) [13]. Fears related to the possibility of transferring the virus to loved ones can thereby restrict activities outside the home not only in those with increased susceptibility for the virus but also in healthy family members. This study also found that these worries resulted in increased stress, which is a common barrier to PA participation. These findings highlight the fact that the presence of PA during pandemic time is a multi-factorial issue that requires further research. 

Research has shown that the presence of social support may have a positive effect on the frequency of PA [14,15]. As such, group PA sessions and use of exercise facilities, parks, and trails for exercise are increasing in popularity. However, considering the COVID-19 pandemic, social distancing and facility closures may limit access to these social activities and facilities, which could negatively impact active time with one’s PA peers. 

During the “Stay-at-Home” orders of the COVID-19 pandemic, significant lifestyle changes have created a “new normal”, but it is not known how this lockdown has impacted the capacity and motivation to be physically active. The primary purpose of this study was to compare the self-reported volume of work, recreation, and travel-related PA, the mode of PA, and the amount of time spent in sedentary behaviors (SBs) prior to and during the COVID-19 “Stay-at-Home” restrictions across North America. A secondary purpose of the study was to assess the impact of personal protection equipment (PPE) requirements on PA participation. Understanding the impact of pandemic restrictions can aid in future programming and intervention for promoting adequate PA during such times. 

## 2. Materials and Methods

### 2.1. Population

Invitations to participate in this online survey were distributed through email and social media across the United States and Canada. The flyer contained a QR Code and website address link to the anonymous Qualtrics survey (Qualtrics XM Research Suite software, 2020 Qualtrics^®^). The first page of the survey contained the informed consent and some general screening questions to determine the eligibility of the participant: adult (≥18 years) from either country, English-speaking, and physically capable of exercise. If the participant was deemed ineligible or did not wish to participate, the survey ended. The study was approved by Ohio University’s Institutional Review Board and the online survey was open between 21 April and 9 May 2020.

### 2.2. Questionnaire

Participants completed demographic questions (age-group, race, sex, education status, city, state, and country) as well as previous (before COVID-19 restrictions) and current (during COVID-19 restrictions) employment status (e.g., employed outside of the home, inside the home, or unemployed). Self-reported moderate and vigorous PA and SB were reported using the Global Physical Activity Questionnaire (GPAQ) capturing data related to work, travel, and recreation [16]. The GPAC questionnaire is a valid (Spearman’s rho ranging from 0.45 to 0.65) and reliable tool (Kappa statistic ranged from 0.67 to 0.73; agreement ranging from 85.6 to 92.1%) for measuring PA in large populations across various age groups and countries [17]. For this survey, work was defined as something you must do, such as “paid employment, studying/training, household chores, farm work, fishing or hunting for food and self-employment”, whereas travel was defined as the usual way of traveling to and from places, such as going to “work, shopping, or place of worship”, and recreation was defined as things you wish to do but do not have to, such as “sports, fitness or recreational (leisure) activities”. The participant was asked to answer these questions related to their normal daily routines (before COVID-19 restrictions) and related to their current situation under COVID-19 restrictions.

At the end of the GPAQ survey, COVID-19-specific questions were included to obtain more information about changes in PA locations, modes, barriers, and PPE use. Using state- or province-specific health department and national public radio websites, the status of the COVID-19 restrictions was determined for each state and province for this two-week period. 

### 2.3. Data Processing and Analyses 

All records of those who declined participation in the survey or those who did not complete all the demographics, employment and GPAQ questions were removed. Proportions were then calculated for the general characteristics of the sample overall and by groups (country, sex, age, race, income, and education). Means (±SD) were calculated for moderate-to-vigorous PA (MVPA) and SB measures overall and by group at each time point separately. Repeated measures one- and two-way ANOVAs with Tukey adjustment were used to determine differences in MVPA and SB (watching TV and sitting) by time point overall and by group with group by time point interactions. Two-way ANOVA (i.e., employment change × time) was used to assess the impact of changes in employment status (i.e., lost or gained employment) or environment (i.e., working outside to inside the home) on MVPA and SB. The COVID-19-related data were reported as frequencies (n, %) to determine the impact of the CDC recommendations and “Stay-at-Home” on MVPA and SB (e.g., MVPA mode, facility use, etc.). Tukey adjustment was applied to account for within-subject similarities where necessary. Required sample size was calculated using the outcome of a recent publication reporting the global impact of COVID-19 on PA [18]. With a 1.22 odds ratio at an alpha of 0.05 and 80% power and an expected 40% of the population changing because of the pandemic, 341 participants were required to detect significance. All analyses were performed in SAS statistical software (version 7.15, SAS Institute, Cary, NS, USA) with an alpha of 0.05 set for significance.

## 3. Results

Within the two weeks of circulating the survey throughout social media and mass emails across the United States and Canada, 1616 people responded to the survey. Of those, 11 (0.7%) were removed due to foreign addresses, 9 (0.6%) declined to complete the survey, and 260 (16.1%) did not complete some portion of the demographics, employment, or work-related MVPA data to be included in the analyses, resulting in 1336 participants (91.9% from the United States; 74.2% female, 88.8% Caucasian; 47.5% between 18 and 29 years) included in all data analyses. Data represented 45 of the United States, including the District of Columbia, Hawaii, and Puerto Rico (missing: Alaska, Connecticut, Nebraska, Rhode Island, South Dakota, Vermont, and Wyoming), and 4 of the Canadian provinces (Alberta, British Columbia, Ontario, and Quebec). Through online investigations of government websites, it was found that 95.9% (*n* = 47 of the 49 regions) were experiencing either a Public Health Emergency, State of Emergency, or “Stay-at-Home” orders at the onset of data collection and all schools remained closed for the duration of data collection in all regions. Some restrictions had begun to lift in 14 of the 45 represented United States, while no restrictions were lifted in the represented Canadian provinces during this same time. It should also be noted that fitness centers/facilities began opening in 7 of the United States by the end of this 2-week period. 

Prior to COVID-19 restrictions, 82.4% (*n* = 1101) reported working outside of the home, which decreased by 61.8% during the restrictions, resulting in only 20.6% (*n* = 275) reporting to continue working outside the home during the COVID-19 restrictions. More specifically, 51.4% (*n* = 687) of the sample reported shifting to working from inside the home, while only 0.4% (*n* = 5) shifted in the opposite direction to begin working outside the home. An additional 12.1% (*n* = 162) of the sample reported losing their current job, while 1.1% (*n* = 15) gained employment from pre- to mid-COVID-19. During this same time period, unemployment in the United States rose from 4.4 to 14.7% from pre- to mid-COVID-19 restrictions [19]. This is comparable to the reported 13.0% unemployment rate in Canada due to COVID-19, up from 7.8% pre-COVID-19 [20]. Demographic information of the study sample overall and by group is reported in Table 1. 

### 3.1. Work-Related MVPA

In assessing the overall changes in reported work-related MVPA from pre- to mid-COVID-19, 8.3% (from 44.2 to 35.9%, respectively) fewer participants reported participating in work-related MVPA mid-COVID-19. Overall, there was a decrease of 178.6 ± 20.9 min/week reported time spent in work-related MVPA (adj. *p* < 0.0001; Figure 1). Since a smaller sample was collected from Canada than from the United States, comparison by country was not possible. In Canada, the impact of COVID-19 on PA was not statistically significant with a decrease in reported time spent in work-related MVPA from 241.8 ± 49.5 to 132.4 ± 35.8 min/week (adj. *p* = 0.08). In contrast, the United States saw a decrease from 420.0 ± 24.4 to 235.3 ± 17.2 min/week in work-related MVPA (adj. *p* < 0.0001). A sex by time interaction was observed in that males reported less of a decrease in time spent in work-related MVPA from pre- to mid-COVID-19 compared to females (adj. *p* values < 0.0001; Figure 1). An age by time interaction was also observed (adj. *p* < 0.0001), which was driven by a highly significant decrease in the younger age groups (18–29 years and 30–39 years), with no significant decrease in the older age groups (60–60 years and 70+ years; Figure 2). When analyzing across race, there was only a significant decrease in work-related MVPA in the Caucasian group (from 400.0 ± 24.0 to 227.2 ± 17.3 min/week; adj. *p* < 0.0001), compared to the African American group (from 554.5 ± 160.0 to 191.8 ± 60.5 min/week; adj. *p* = 0.17), Asian group (from 437.5 ± 141.6 to 267.6 ± 96.7 min/week; adj. *p* = 0.91), and Hispanic group (from 416.5 ± 126.9 to 275.0 ± 102.0 min/week; adj. *p* = 0.73). The sample of Indian/Alaskan was too small (*n* = 3) to include in this analysis. Differences in income brackets did not affect the change in reported work-related MVPA (adj *p* values < 0.01), nor did lower education levels (high school and Associates degree; adj. *p* > 0.05). When comparing the changes in work-related MVPA in those who experienced a change in employment status (lost employment) or environment (shifted to working from home) to those with no change, a change in employment status resulted in a decrease in MVPA from 766.5 ± 87.4 to 155.5 ± 32.6 min/week and a change in work environment resulted in a decrease in MVPA from 268.4 ± 25.5 to 112.3 ± 13.8 min/week (adj. *p* values < 0.0001), compared to no difference in those who had no change in employment environment (from 461.0 ± 39.3 to 393.4 ± 35.9 min/week; adj. *p* = 0.11; Figure 3).

### 3.2. Travel-Related MVPA

In assessing the overall changes in reported time spent in travel-related MVPA from pre- to mid-COVID-19, 28.0% (from 54.3 to 26.3%), fewer participants reported participating in travel-related MVPA mid-COVID-19. There was a similar decrease in reported time spent in travel MVPA between countries (Canada: −152.6 ± 14.1 vs. USA: −188.1 ± 51.9 min/week; adj. *p* values < 0.0001). A sex by time interaction was observed in that males reported less of a decrease in time spent in travel-related MVPA from pre- to mid-COVID-19 compared to females (adj. *p* values < 0.0001; Figure 1). An age by time interaction was observed for travel-related MPVA (adj. *p* < 0.0001) with a decrease across the younger age groups, with no significant decrease in the older age groups (60–69 years and 70+ years; Figure 2). When analyzing across race, there was a significant decrease in the reported time spent in travel-related MVPA in the Caucasian (from 279.8 ± 16.9 to 110.5 ± 11.3 min/week; adj. *p* < 0.0001) and Hispanic (from 314.2 ± 59.4 to 77.1 ± 24.2 min/week; adj. *p* = 0.003) groups, compared to the African American (from 230.0 ± 72.8 to 76.7 ± 19.7 min/week; adj. *p* = 0.99) and Asian (from 184.9 ± 59.4 to 79.9 ± 30.9 min/week; adj. *p* = 0.25) groups. Time spent in reported travel-related MVPA differed from pre- to mid-COVID-19 across all income brackets (ranging from −179.3 ± 37.3 to −349.3 ± 82.5 min/week), in only those with at least some college education or higher (ranging from −117.2 ± 89.2 to −306.1 ± 42.1 min/week), and regardless of changes in employment status and environment (ranging from −173.4 ± 30.7 to −329.0 ± 63.1 min/week; adj *p* values < 0.0001).

### 3.3. Recreational MVPA

In assessing the overall changes in reported recreational MVPA participation from pre- to mid-COVID−19, 7.3% (from 87.0 to 79.8%, respectively) fewer participants reported participating in recreational MVPA mid-COVID-19. In contrast to work and travel MVPA, there was no significant change in time spent in recreational MVPA in the entire sample in response to COVID-19 restrictions (−30.4 ± 11.5 min/week; adj. *p* = 0.69), with an overall mean of 506.7 ± 23.6 min/week. When recreational MVPA was examined separately by intensity, there was a slight but significant decrease in moderate (282.5 ± 15.4 to 243.3 ± 13.4 min/week; adj. *p* < 0.0001) and vigorous (231.7 ± 13.2 to 197.0 ± 13.2 min/week; adj. *p* < 0.0001) intensity PA from pre- to mid-COVID-19. However, the lack of significant change in reported time spent in recreational MVPA was consistent across all groups (by country, sex, age, race, income, education, etc.; adj. *p* values ranged from 0.69 to 1.00). Consequently, recreational MVPA was not included in Figure 1, Figure 2 and Figure 3.

### 3.4. Sedentary Behavior

In assessing the changes of self-reported behaviors from pre- to mid-COVID-19 restrictions, there was a significant increase in time spent in SB (+94.9 ± 4.1 min/week and +108.3 ± 4.9 min/week of TV viewing and sitting, respectively) when assessing the entire sample across time points. There was a similar increase in reported time spent watching TV (Can: +86.3 ± 10.6 and USA: +95.6 ± 4.4 min/week; adj. *p* values < 0.0001) and sitting (Can: +112.8 ± 14.6 and USA: 107.9 ± 5.2 min/week; adj. *p* values < 0.001) between the two countries. A sex-by-time interaction was observed in that males reported less of an increase in time spent watching TV and sitting compared to females (Figure 1). Time spent watching TV and sitting significantly increased for all ages, except the oldest age group (70+ years; Figure 2). When analyzing across race, there was a significant increase in reported time spent watching TV in all racial groups: Caucasian (167.0 ± 4.5 to 288.1 ± 6.9 min/week; adj. *p* < 0.0001); African American (167.3 ± 21.9 to 348.3 ± 44.3 min/week; adj. *p* < 0.0001); Asian (188.3 ± 33.6 to 279.4 ± 34.5 min/week; adj. *p* < 0.001); and Hispanic (152.5 ± 16.3 to 291.3 ± 45.2 min/week; adj. *p* < 0.001) groups. Whereas there was a significant increase in reported time spent sitting in the Caucasian (406.7 ± 46.1 to 545.4 ± 8.5 min/week; adj. *p* < 0.0001) and African American (405.0 ± 46.1 to 616.0 ± 65.7 min/week; adj. *p* < 0.0001) groups but not the Asian (399.4 ± 43.0 to 492.2 ± 45.9 min/week; adj. *p* = 0.06) and Hispanic (437.5 ± 48.9 to 545.0 ± 35.3 min/week; adj. *p* = 0.11) groups. The change in SB did not differ across income brackets (adj. *p* values < 0.0001), education levels (adj. *p* values < 0.05), or changes in employment (adj. *p* values < 0.0001).

### 3.5. Physical Activity Mode

In the overall sample, 1029 (77.0%) of the participants reported participating in some mode of PA before the COVID-19 restrictions. The predominant PA mode consisted of locomotive type activities (*n* = 983, 73.6%), such as walking, jogging, skating, and hiking, and fitness classes, such as Zumba, yoga, or dance (*n* = 499, 37.4%). Mid-COVID-19, 38.2% (*n* = 510) of the sample reported a change in PA mode in response to the “Stay-at-home” restrictions, whereas 38.8% (*n* = 519) reported no change. While overall participation in locomotive and fitness class PA types declined to 8.6% and 8.4%, respectively, they remained the predominant PA modes mid-COVID-19. Weight room or gym-type activities (i.e., resistance training, calisthenics, HIIT, and CrossFit) declined from 35.6 to 5.1%, whereas riding type activities (i.e., cycling, rowing, and horseback) declined from 18.7% to 2.0% from pre- to mid-COVID-19. Similarly, equipment use declined from 28.7 to 1.8% from pre- to mid-COVID-19. Activities, such as climbing, golf, swimming, and sports disappeared almost completely (<1%) in response to the COVID-19 restrictions. In contrast, there was an increased reliance on virtual media (i.e., online fitness classes, active video gaming, etc.) for PA participation. Pre-COVID-19, only 0.8% (*n* = 11) of the sample reported using virtual media for PA, which increased to 6.7% (*n* = 89) mid-COVID-19.

### 3.6. Facility Use

Before “Stay-at-Home” orders were in place, 68.5% (*n* = 709) of our sample used gym facilities at least occasionally to maintain their PA. When asked if they would return to these facilities when they re-opened, the responses were mixed, with 35.6% (*n* = 372) saying they would return, 33.1% (*n* = 337) saying they were unsure, and the balance saying they would not. Reasons participants stated they would not return to the gym were because they did not feel safe (52.7%, *n* = 532), they felt their exercise outside the gym was sufficient (33.2%, *n* = 330), or they did not feel gyms were essential businesses and should remain closed (27%, *n* = 268). Participants stated they would return to the gym because they would get a better workout (34.8%, *n* = 346), they needed a place to exercise (28.8%, *n* = 286), they were tired of exercising at home (22.1%, *n* = 219), and they felt the facilities were safe (15.6%, *n* = 155).

### 3.7. PPE Use

When asked about how often they used PPE, such as face coverings or gloves, during PA outdoors before the COVID-19 pandemic and restrictions, 65.3% (*n* = 655) of our sample said they never used PPE, while 15% (*n* = 155) used it sometimes, 4.8% (*n* = 50) used it half the time, and 8% used it most of the time (*n* = 84) or always (*n* = 82). Most participants (68%, *n* = 705) said they were not using PPE during the pandemic, primarily because they did not think it was necessary (100%) or found it uncomfortable (28.5%, *n* = 201). An equal number of participants reported not wearing PPE because of looks or because it was unavailable to them (18.3%, *n* = 129). Of those who were using PPE during exercise during the pandemic, the majority reported wearing a homemade mask (27.9%, *n* = 287), while 10.3% (*n* = 106) wore a disposable mask; 8.4% (*n* = 83) wore a bandana, balaclava, or ski mask; 5.1% (*n* = 53) wore an N95 mask; and only 8.3% (*n* = 85) reported wearing gloves.

## 4. Discussion

The purpose of this study was to assess changes in PA and SB in response to COVID-19 “Stay-at-Home” restrictions in the United States and Canada. The restrictions included a shift from working outside the home to working inside the home for all non-essential personnel and the closure of many businesses, including fitness centers and recreation trails. Perception of mitigation strategies, such as “Stay-at-Home” orders, was high amongst US residents (79.5%) [21], while half the population surveyed in Canada believed restrictions went far enough; although, in some provinces, as many as 40% believed their government did not go far enough [22]. 

Although necessary, social isolation to prevent the spread of the virus established a new social dynamic that resulted in profound and undesirable changes in people’s lifestyles. The time window created by a less busy life was not translated into higher levels of PA, as participation in PA decreased in general. Physical inactivity and SB so present and rooted in the world already were further exacerbated by the COVID-19 pandemic, establishing a new lifestyle routine, even more harmful to health. This, in turn, could lead to the development of chronic diseases or the worsening of pre-existing diseases that are related to PA and SB, such as obesity, diabetes, and hypertension [3]. Sallis and Pratt (2020) expanded on these benefits of PA participation during a pandemic, providing evidence that PA participation can help the immune system fight off viral infection, reduce the symptom severity of the coronavirus, reducing COVID-related stress, protect the lungs from severe COVID-related respiratory symptoms, and may enhance the antibody response to vaccinations [23]. In our study, we found that a change in employment environment and status both had a negative impact on PA and SB, regardless of income or education level. This impact was worse in women, younger adults, and Caucasian participants. On the contrary, men, older age groups, and some races have not been affected as much by these restrictions. This information is valuable for directing our attention for developing strategies to improve these more sedentary, less active lifestyles. 

While there is no way to directly relate changes in behavior to the COVID-19 “Stay-at-Home” restrictions, the data clearly indicated a reduction in time spent in PA and an increase in time spent in SB compared to pre-pandemic values. This study separated PA participation into work, recreational, and travel, finding no difference in recreational PA during COVID-19 restrictions compared to pre-pandemic reported values. However, there was an 8.3% reduction in reported work-related PA, with greater reductions in females, younger age groups, and Caucasians. This reduction was also more pronounced in those who either lost their jobs (−611 min/week) or shifted to working from home (−156.1 min/week) compared to those with no changes in employment status or environment. Like these findings, a study by Spence et al. (2021) including both Canada and the United States found that 55% of the sample self-reported less work-related PA (*N* = 1521; 30–64 y) using a 4-point Likert scale for answering a two-item measure to assess typical PA during the three months prior to COVID-19 outbreak compared to during COVID-19 lockdown [24]. Contrary to the current study, which found no difference in SB related to changes in employment, Spence et al. (2021) found an 84% increase in work-related screen time, an indicator of SB. With respect to travel-related PA participation, the current study also reported a 28% drop in MVPA, which was less than what was found in Spence et al. (2021). They reported a 58% reduction in PA for travelling to and from work or school. This was complimented by 81% of the sample participating in PA in or around the home and 63% in their neighborhood as opposed to travel-related PA. 

Another study by Silva et al. (2021) used a similar online or phone questionnaire (International Physical Activity Questionnaire) in Portuguese adults (*N* = 5856; ≥16 years) to assess PA habits in response to COVID-19 social confinement [25]. Approximately 54% of the sample did not meet the PA recommendation during social confinement, more so in the lower education levels (55.4%) compared to those with at least a university degree (47.8%). This contradicts the current study that found the COVID-19 restrictions did not affect work-related MVPA in those with lower education levels (high school or associates degree; adj. *p* > 0.05) compared to significant reductions in those with bachelor, master, or doctoral degrees. The reverse was observed in travel-related MVPA.

Generally, the engagement in PA is an individual choice. However, the mandatory restrictions imposed by COVID-19, which included the closing of public parks, gyms, and sports facilities, reduced the autonomy of people to choose their PA mode. According to the social-cognitive theory, increased perceived barriers to PA, as in the restricted access to fitness facilities, can negatively impact PA participation [4]. While over half of the sample used gyms regularly, there was no consensus as to whether they would return to using these gyms once they opened, stating feeling unsafe as the main reason. According to the theory of planned behavior, there may have been a shift in the risk/benefit ratio with respect to PA participation in such facilities [5,7]. Prior to COVID-19, it was likely perceived that PA participation would improve overall health and well-being. However, the threat of COVID-19 exposure while using these facilities or during large group activities could be perceived to overshadow these health benefits. As seen in Mertens et al. (2020), the fear of transmitting this virus to loved ones was the most common reason for self-isolation and a reduction in PA participation in fitness facilities [13]. With the closing of these facilities the mode of exercise switched to walking and virtual classes or gaming; however, the shift in PA mode did not entirely replace the volume of pre-COVID-19 PA. These findings are like that of Faulkner et al. (2021), who reported greater participation in walking and running, gardening, and online fitness classes and a reduction in sporting activities, resistance training, and in-person fitness classes across several countries [26]. While they do not compare to pre-pandemic values, Ding et al. (2021) also reported that 61.4% of the sample across 11 countries (*N* = 8995; ≥18 years) participated in home-based PA and of them, 38.5% reported participating in PA indoors only [18]. Further, they found that 91.9% of the sample participated in PA either alone or with family members and 63.8% used online fitness classes/programs. Public policies should encourage adherence to practices that are available to everyone, such as exercising at home (i.e., weightlifting, gardening, stair climbing, or choosing a standing posture in the home office). These strategies are necessary since there is no deadline for the end of this pandemic and different countries have already experienced a second or even a third wave of the mass infection. Furthermore, it would be prudent for employers to adopt employee health programs that provide on-demand type physical activity, which can be completed virtually or from home. These types of programs are successful at improving cardiopulmonary and skeletal muscle function in a wide population of adults and would likely benefit whole families.

While access to facilities is a common perceived barrier for PA, PPE requirements can also be seen as a barrier. Mask mandates were in place in only 14 states or locations (including Puerto Rico and Washington D.C) during the data collection period for our survey (21 April–9 May 2020). Of those, only four required masks in all public locations, three required them at business and on public transportation, and the remaining seven required them in public when social distancing could not be maintained [27]. In a survey of 1141 US adults in July 2020, 89.9% of the participants indicated they wore a mask when going into public in response to these mandates [28]. However, less than 50% said they would wear one in a public park or beach. Interestingly, over half the participants said they did not intend to go back to a gym or fitness facility, which is higher than we report in our sample. Of those that said they would go back, 25% said they would be extremely likely to wear a mask when doing so. The CDC considers wearing a mask and physical distancing important during exercise [29]. Mask wearing has the potential to decrease viral spread through aerosol droplets, which is likely to be exacerbated during exercise [30]. Wearing a mask during exercise can potentially cause discomfort, especially if they restrict airflow (i.e., when wet from sweat), creating micro-hypoxic environments [31], which may result in dizziness. The CDC’s recommendation is for individuals to have multiple masks available to them and change the mask when it becomes wet (CDC website). For healthy people, wearing masks during exercise has not been shown to be harmful. However, individuals affected by lung disease, such as asthma or COPD, or heart disease, should be evaluated by a healthcare provider before attempting exercise with any mask. It is further recommended by the CDC that if the intensity of the exercise makes it difficult to wear a mask, it is especially important to do those activities outdoors away from others. Most participants in our study were not using PPE during PA, primarily because they did not believe it was necessary (i.e., not exercising indoors or near other individuals), or because it was uncomfortable. There was no indication that mask wearing was perceived as a barrier to participating in PA. 

## 5. Limitations

The main limitation of the study is represented by the self-reported questionnaire that asked the participant to provide information about their pre- and mid-COVID-19 behaviors at the single time point. While this is considered a convenient sample, the strengths of the study include a wide geographic representation throughout the United States and the period in which it took place, during the early-COVID-19 period. However, the small samples, especially from Canada, could not be considered generalizable. Another potential limitation is that this survey was conducted near the start of the pandemic and “Stay-at-Home” restrictions. We are a year later and do not know how these behaviors have changed. There is now a greater experience with masks and mask wearing, as well as living under (mostly lightened) restrictions. A strength of this study is the high rate of completed or valid responses, which could be attributed to many factors, including the timeliness of data collection. Data was collected not long after “Stay-at-Home” orders began, early into the pandemic and before the possibility of survey fatigue on this topic. However, other papers on this topic have published similar completion rates (75–80%) [18,32].

## 6. Conclusions

In conclusion, our results indicate a negative impact of the “Stay-at-Home” restrictions on PA and SB, in both Canada and the United States. The results are even worse in the specific groups composed of women, younger adults, and Caucasian people. These observations are crucial as the world was already facing a pandemic before COVID-19: the physical inactivity pandemic. The social distancing strategies to mitigate the COVID-19 spread seem to be sinking us further into an inactive and sedentary routine that can become the “new normal”, even after activities resume and COVID-19 is no longer a problem. Thus, understanding the effect of COVID-19 on the population’s PA and SB is a key factor to guide consistent public policies to promote healthy habits, which can have greater adherence in times of health crisis, since the world’s attention is focused on improving health.

## Figures and Tables

**Figure 1 ijerph-18-11935-f001:**
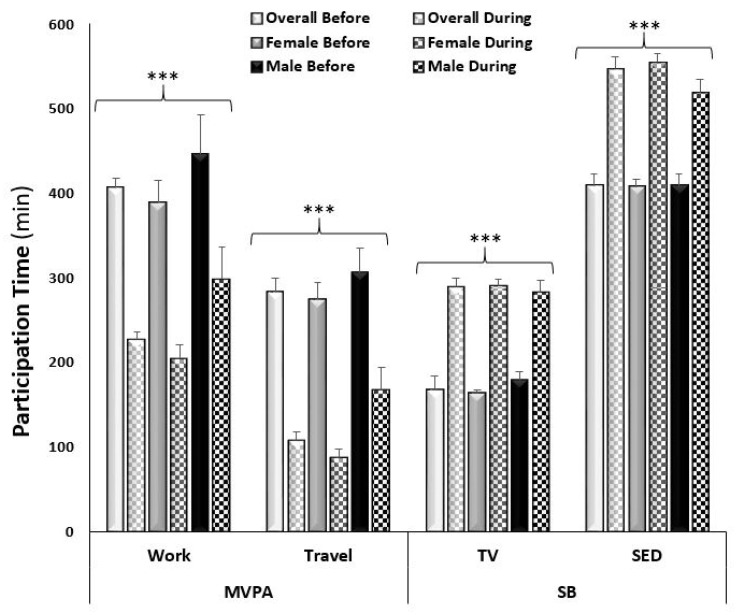
Average weekly self-reported time (minutes) spent in moderate-to-vigorous physical activity (MVPA) related to work and travel and time spent in sedentary behaviors (watching TV and sitting). *** Significant difference by time point overall and a significant sex by time point interaction (adj. *p* < 0.0001).

**Figure 2 ijerph-18-11935-f002:**
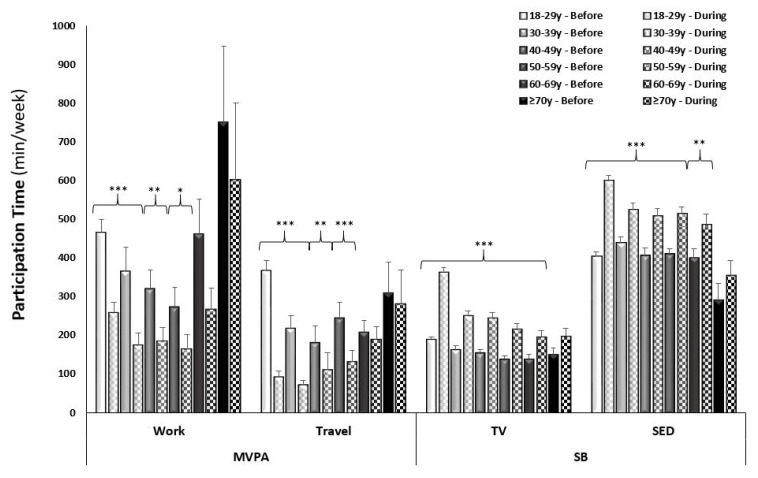
Average weekly self-reported time (minutes) spent in moderate-to-vigorous physical activity (MVPA) related to work and travel and time spent in sedentary behaviors (watching TV and sitting) separated by age group. Significant difference within age group by time point (*** adj. *p* < 0.0001; ** adj. *p* < 0.001; * adj. *p* < 0.05).

**Figure 3 ijerph-18-11935-f003:**
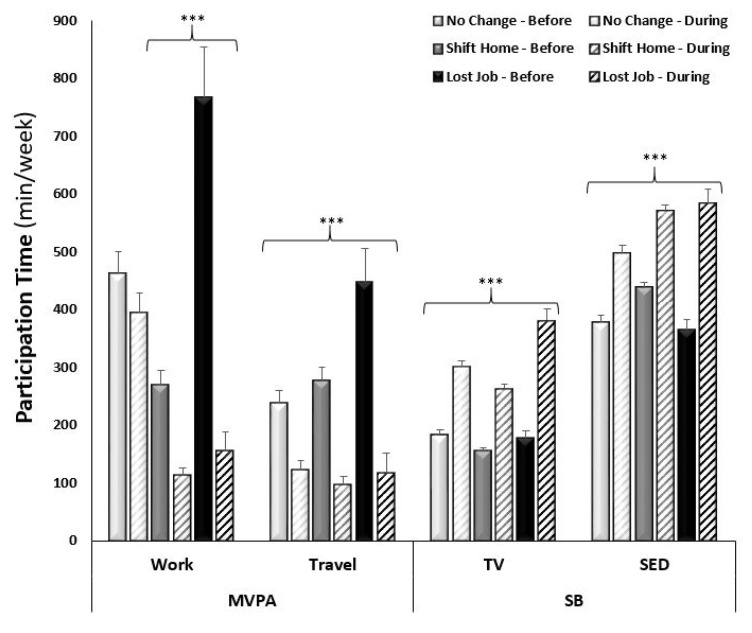
Average weekly self-reported time (minutes) spent in moderate-to-vigorous physical activity (MVPA) related to work and travel and time spent in sedentary behaviors (watching TV and sitting) separated by employment status or environment change by time point. Significant difference within employment status and environment change by time point (*** adj. *p* < 0.0001).

**Table 1 ijerph-18-11935-t001:** Participant Characteristics (*n*, %).

	Frequency	Percent		Frequency	Percent
**Country**			**Race/Ethnicity**		
United States	1228	91.9	White	1187	88.8
Canada	108	8.1	Black	46	3.4
**Sex**			Asian	40	3.0
Female	991	74.2	Am. Indian/Alaskan	3	0.2
Male	335	25.1	Hispanic/Latino	39	2.9
Prefer Not to Say	10	0.7	Prefer Not to Say	21	1.6
**Age Group**			**Reported Income**		
18–29	634	47.5	<25,000	243	18.2
30–39	228	17.1	25,000–50,000	157	11.8
40–49	201	15.0	50,000–75,000	211	15.8
50–59	170	12.7	>75,000	168	12.6
60–69	84	6.3	>100,000	431	32.3
≥70	19	1.4	Prefer Not to Say	126	9.4
**Level of Education**			**Residence Type**		
High School	27	2.0	Single-Family Home	1002	75.0
Some College	290	21.7	Condominium	43	3.2
Associates Degree	81	6.1	Duplex	32	2.4
Bachelor’s Degree	357	26.7	Apartment	237	17.7
Master’s Degree	344	25.7	College Dorm	6	0.4
Doctoral Degree	237	17.7	Other	16	1.2
**Employment Environment and Status**					
Shifted to working from home				687	51.4
Shifted to working outside the home				5	0.4
Lost employment				162	12.1
Gained employment				15	1.1
No change				467	35.0

Total sample size, *N* = 1336.

## Data Availability

Data set is available upon request at DOI: 10.7303/syn26454110.

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
