# Peer review of "Impact of COVID-19 Stay-at-Home Restrictions on Employment Status, Physical Activity, and Sedentary Behavior"

_ijerph, 2021, doi:10.3390/ijerph182211935_

Round 1

Reviewer 1 Report

I read with great interest the work entitled “ Impact of COVID-19 Stay-at-Home Restrictions on Employment Status, Physical Activity and Sedentary Behavior”

The  study compared self-reported MVPA and SB before and during COVID-19 “Stay-at-home” restrictions as a potential barrier across North America.

I ask the authors some small but fundamental clarifications to make the paper publishable.

Major Concerns

1) The introduction needs to be expanded. In particular, it is suggested to point that restrictive measures, especially the strict ones, can also cause problems due to staying in the same environment and increasing the number of hours in front of smartphones, PCs or TVs.

Please cite: The “quarantine dry eye”: the lockdown for coronavirus disease 2019 and its implications for ocular surface health. Risk management and healthcare policy, 14, 1629.

It should be introduced that the fear of COVID-19 has reduced, in addition to physical activity, access to check-ups (even those for emergency situations). Lack of physical activity and health checks can have devastating effects.

Please cite: Fear of the COVID-19 and medical liability. Insights from a series of 130 consecutives medico-legal claims evaluated in a single institution during SARS-CoV-2-related pandemic.

2) Regarding the methods, I ask the authors if the questionnaire has been validated and if the estimate of the sample size has been made before starting the study.
3) Discussions are mainly focused on the use of PPE. I think it is necessary to broaden it by also commenting on the other results. For example, fewer minutes of activity are matched with weight gain?
4) For the rest I would ask for the limitations to be put in a separate chapter in the discussions.

5) Finally, I would ask to increase the number of references which I consider insufficient.

Author Response

Thank you for your time and efforts in reviewing this manuscript. Your comments and suggestions have greatly improved the quality of this paper. Please find details responses to each of your comments attached. 

Reviewer 2 Report

Although facility closures might limit access to social activities, individual physical activity such as cycling and running became popular during the COVID-19 pandemic. Therefore, the theoretical framework of this study needs to be re-established.

Please elaborate on how the COVID-19 pandemic influenced people’s lives.

Please report the validity of the questionnaire.

Response rate: 83% (1336/1616). Compared with other online survey studies, the response rate of this study was significantly high. Please elaborate on how to obtain a high response rate.

The Results section is thin. This is a cross-sectional study. Merely comparing differences is not enough.

Author Response

(The authors gave the same response as above.)

Round 2

Reviewer 1 Report

I read with pleasure the new version of the paper entitled “Impact of COVID-19 Stay-at-Home Restrictions on Employment Status, Physical Activity and Sedentary Behavior”.

The suggestions from the previous round were mostly satisfactorily addressed.

The only note I would like to make to the authors concerns the citation of the article "Fear of COVID-19 and Medical Liability. Insights from a series of 130 consecutive medico-legal complaints evaluated in a single institution during the SARS-CoV-2 pandemic. The meaning, however correct, given by the authors to the suggestion should be broadened in a concise way.

The fear of COVID-19 in addition to the sense of fear and anxiety in the population has led to less access to specialists (ie nutritionists, endocrinologists, psychiatrists) who, by intervening in time, could have mitigated the effects of the decreased activity that have been well described by the authors.

I think this small addition is important which, with the nuance and the quotation indicated, could further improve the work .

I have enjoyed the paper globally and hope that the correction can be made in a short time.

Author Response

Thank you for your kind comments and for your feedback. 

Point 1: The only note I would like to make to the authors concerns the citation of the article "Fear of COVID-19 and Medical Liability. Insights from a series of 130 consecutive medico-legal complaints evaluated in a single institution during the SARS-CoV-2 pandemic. The meaning, however correct, given by the authors to the suggestion should be broadened in a concise way.

The fear of COVID-19 in addition to the sense of fear and anxiety in the population has led to less access to specialists (i.e., nutritionists, endocrinologists, psychiatrists) who, by intervening in time, could have mitigated the effects of the decreased activity that have been well described by the authors.

I think this small addition is important which, with the nuance and the quotation indicated, could further improve the work .

Response 1: While we appreciate the feedback from the reviewer, the authors respectively disagree with this comment. The primarily role of these specialists (i.e., nutritionists, endocrinologists, psychiatrists) is not to address physical activity, as most physicians are not trained in exercise interventions or prescription and do not provide these services to their patients. We offer this reference in support of our response: 

    • Okafor and Goon, 2021. Physical Activity Advice and Counselling by Healthcare Providers: a scoping review. Healthcare (Basel). May 19;9(5):609.  doi: 10.3390/healthcare9050609.

Reviewer 2 Report

The statements of the authors are contradictory. For example, the authors state that “The primary purpose of this study was to compare the self-reported volume of work, recreation and travel-related PA, the mode of PA, and the amount of time spent in sedentary behaviors (SB) prior to and during the COVID-19 ‘Stay-at-Home’ restrictions across North America. (p.3)” However, “the study was approved by Ohio University’s Institutional Review Board and the online survey was open between April 21 – May 9, 2020. (p.3)” In other words, the authors did not collect data before the COVID-19 pandemic. How do the authors compare data prior to and during the COVID-19 pandemic?

In addition, the authors state that “A secondary purpose of the study was to assess the impact of personal protection equipment (PPE) requirements on PA participation. (p.3)” However, the authors do not specify the relationship between PPE requirements and PA participation. A clearer study framework is needed.

Author Response

Point 1: “How do the authors compare data prior to and during the COVID-19 pandemic?”

Response 1: We apologize for any confusion on this point. We asked questions related to how activity levels changed from before to during the stay-at-home orders (as explained in the methods: “The participant was asked to answer these questions related to their normal daily routines (before COVID-19 restrictions) and related to their current situation under COVID-19 restrictions. At the end of the GPAQ survey, COVID-19-specific questions were included to obtain more information about changes in PA locations, modes, barriers, and PPE use”

Point 2: A secondary purpose of the study was to assess the impact of personal protection equipment (PPE) requirements on PA participation. (p.3)” However, the authors do not specify the relationship between PPE requirements and PA participation.

Response 2: We apologize for any confusion on this point. We were not assessing the relationship between PPE requirements and PA participation, we simply wanted to know if wearing PPE during PA was occurring and/or if it was a barrier. Therefore, participants were asked about their PPE use before and during COVID-19 restrictions and we specifically asked participants if wearing PPE during PA and if not, why. Those data are presented in the results and discussed in the paper.

Point 3: As for the first reviewer’s comments that “specialists (i.e., nutritionists, endocrinologists, psychiatrists) who, by intervening in time, could have mitigated the effects of the decreased activity that have been well described by the authors.”

Response 3: Thank you for your comment but we respectively disagree. The primarily role of these specialists is not to address physical activity, as most physicians, nutritionists and psychiatrists are not trained in exercise interventions or prescription and do not provide these services to their patients. We cite the following reference for support of this statement: Okafor and Goon 2021. Physical Activity Advice and Counselling by Healthcare Providers: a Scoping Review. Healthcare (Basel), May 19; 9(5): 609. doi: 10.3390/healthcare9050609